# Retrieval of Prefabricated Zirconia Crowns with Er,Cr:YSGG Laser from Primary and Permanent Molars

**DOI:** 10.3390/ma13235569

**Published:** 2020-12-07

**Authors:** Janina Golob Deeb, Connor McCall, Caroline K. Carrico, William O. Dahlke, Kinga Grzech-Leśniak

**Affiliations:** 1Department of Periodontics, School of Dentistry, Virginia Commonwealth University, Richmond, VA 23284, USA; jgolobdeeb@vcu.edu; 2Department of Pediatric Dentistry, School of Dentistry, Virginia Commonwealth University, Richmond, VA 23284, USA; mccallc@mymail.vcu.edu (C.M.); wodahlke@vcu.edu (W.O.D.); 3Dental Public Health and Policy, School of Dentistry, Virginia Commonwealth University, Richmond, VA 23284, USA; ckcarrico@vcu.edu; 4Laser Laboratory, Department of Oral Surgery, Wroclaw Medical University, 50-367 Wroclaw, Poland

**Keywords:** debonding, Er,Cr:YSGG laser, primary teeth, zirconia crown

## Abstract

(1) Background: Prefabricated zirconia crowns are used to restore teeth in children. The purpose of this study was to evaluate the removal of these crowns with the erbium, chromium: yttrium-scandium-gallium-garnet (Er,Cr:YSGG) laser; (2) Methods: A total of 12 primary and 12 permanent teeth were prepared and prefabricated zirconia crowns were passively fitted and cemented with two resin modified glass-ionomer (RMGI) cements. Surface areas of prepared teeth and crowns were calculated. Crowns were removed using two laser settings: 4.5 Watts, 15 Hertz, 20 water/20 air, and 5 Watts, 15 Hertz, 50 water/50 air. The retrieval time and temperature changes were tested recorded. Data were analyzed using ANOVA with Tukey’s adjusted post hoc pairwise comparison *t*-test; (3) Results: The average time for crown removal was: 3 min, 47.7 s for permanent; and 2 min 5 s for primary teeth. The mean temperature changes were 2.48 °C (SD = 1.43) for permanent; and 3.14 °C (SD = 1.88) for primary teeth. The time to debond was significantly positively correlated with tooth inner surface area and volume, outer crown volume, and the cement volume; (4) Conclusions: Use of the Er,Cr:YSGG laser is an effective, safe and non-invasive method to remove prefabricated zirconia crowns cemented with RMGI cements from permanent and primary teeth.

## 1. Introduction

Dental caries continues to be the most common chronic childhood disease [1]. It can greatly affect a child’s well-being; lead to infection, poor nutrition, and missed school days; and negatively influence learning and decrease overall quality of life. Clinical dentistry has evolved to treat this disease by removing the caries and restoring teeth with fillings or crowns.

The stainless-steel crown (SSC) has long been considered the “gold standard” to restore teeth with caries, cervical demineralization, developmental defects, and to temporarily restore permanent molars treated with a root canal therapy in the growing patient [2]. Multiple longitudinal studies have demonstrated the superiority of SSCs over amalgam restorations for primary molars with multi-surface involvement [3,4]. Parent’s increasing desire for esthetic restorations for their children has led to the implementation of prefabricated zirconia crowns to the pediatric dentist’s armamentarium, as an alternative to SSCs. Zirconia is a crystalline dioxide of zirconium with mechanical properties similar to those of metals, and color similar to teeth. Ready-made zirconia crowns are available for primary incisors, molars and permanent molars [5]. Prefabricated zirconia crowns share the same indications for use as do SSCs [6]. While SSCs have better retention, zirconia crowns allow for better gingival health and less plaque accumulation, in addition to vastly better esthetics [6,7]. Zirconia crowns require more circumferential tooth reduction (minimum 1.5–2 mm) for a required passive fit. This passive fit makes strong cementation critical for their longevity [8]. From time to time, a practitioner may find it necessary to remove a previously cemented zirconia crown. Removal can be accomplished by sectioning using rotary instruments, which is time consuming and stressful for the patient and clinician. Sectioning with high-speed burs can lead to iatrogenic tooth damage as it can be difficult to differentiate between the tooth and the tooth-colored materials [9]. Glass ionomer cements have been the standard for SSC cementation as they provide fluoride release and bonding to dentin. Resin-modified glass ionomer (RMGI) cements are used for cementing prefabricated zirconia crowns. They combine the properties of traditional glass ionomers with the handling and light-curing abilities of resin, culminating in increased retention and longevity.

Erbium lasers have been explored for removal of ceramic crowns [10]. The light emitted by erbium lasers is transmitted through the translucent material of the crown and s absorbed by water molecules and residual monomers in the resin and glass ionomer cements. This leads to ablation of the cement resulting in reduced bond strengths and hydrodynamic ejection [11]. This strategy has been successfully applied for debonding of ceramic orthodontic brackets from teeth [12]. Temperature changes in the pulp chamber during laser irradiation should remain within tolerable range to not adversely affect the vitality of the pulp [12,13]. The time required to remove lithium disilicate crowns with high-speed burs takes approximately 6 min compared to laser assisted removal that can be done in 60–90 s [14]. Crowns removed by sectioning are destroyed, while laser debonded crowns remain undamaged and can be recemented.

Erbium family lasers include two lasers with similar properties except slight differences in wavelength range, pulse duration and energy: (1) erbium-doped-yttrium aluminum garnet (Er:YAG) (2940 nm); and (2) erbium, chromium: yttrium-scandium-gallium-garnet (Er,Cr:YSGG) (2780 nm). Recent studies have proven a wide range of dental applications for erbium family lasers including oral pathogen control [15], soft tissue de-epithelization [16], bone decortication [17], and fiber post retrieval [18]. The use of Er:YAG laser for removal of lithium disilicate and zirconia crowns from zirconia and titanium implant abutments has been deemed safe and efficient [19,20,21]. Erbium family lasers have also been used successfully for removal of ceramic crowns from permanent teeth [10,22].

Facilitating crown removal with laser as a non-invasive alternative to rotary instruments could improve clinical practice considering that pre-formed zirconia crowns are becoming more common in the pediatric population. To date, no studies of this type have been conducted on primary teeth.

The aim of this in vitro study was to analyze the feasibility of using an Er,Cr:YSGG laser for retrieval of prefabricated zirconia crowns cemented with resin-modified glass ionomer (RMGI) cements from primary and permanent molars while also establishing most the effective and least aggressive laser settings. Our second aim was to explore if any relationships exist between the time required to remove a crown and the surface area of the crown, abutment tooth and the cement volume. Our third aim was to assess temperature changes in the pulp chamber during laser irradiation at two different settings. Additionally, it was of interest to examine how laser irradiation affects the structure of the crown and tooth surface and if previously performed debonding has an effect on adhesion properties during subsequent recementation.

## 2. Materials and Methods

Extracted teeth were collected in VCU Pediatric Dental and Oral Surgery clinics and subsequently stored in isotonic saline solution. The amount of remaining non-carious tooth structure was evaluated and those teeth with fractured crowns, gross caries, or previous restorations were excluded from the study. The project was reviewed and approved by the Institutional Review Board (HM20019893) as a Not Human Subjects Research study.

A total of 12 primary (G1, N = 12) and 12 permanent teeth (G2, N = 12) were prepared following the crown manufacturer’s instructions (Nusmile^®®^, Houston, TX, USA) with a 1–2 mm occlusal reduction and approximately 20–30% overall clinical crown reduction into dentin using a dental highspeed rotary handpiece and diamond burs. A 368-023 football-shaped coarse diamond bur was used for occlusal reduction (Henry Schein^®^, Melville, NY, USA), the interproximal sites and the entire clinical crowns of the teeth were reduced using a 169 L taper fissure plain carbide bur (Henry Schein^®^). To establish a chamfer/feather-edge margin the 169 L and an 850-010 tapered-diamond bur (Henry Schein^®^) were utilized. Lastly, all the line angles of the preparations were rounded with the tapered-diamond bur and football-shaped coarse diamond bur to remove any sharp angles and provide for a slightly tapered preparation that allowed the zirconia crown to fit passively.

Prepared tooth specimens were scanned using computer assisted digital (CAD) technology with a Planmeca Emerald intraoral scanner (Emerald, Planmeca, Helsinki, Finland). Scan files were imported into Meshmixer© software (SCR_015736, Cura v.4.4, Ultimaker, The Netherlands) to conduct 3D mesh analysis to calculate tooth surface area (mm^2^) and the tooth volume (mm^3^). (Figure 1A) Prefabricated zirconia crowns (Nusmile^®®^, Houston, TX, USA) were selected for the most intimate passive fit determined digitally before cementation, dried and cemented on teeth following manufacturer’s guidelines. Two cements were used in this study. BioCem (BioCem Universal BioActive Cement; NuSmile: Houston, TX, USA) is the manufacturer’s proprietary RMGI cement with bioactive properties including fluoride release and, phosphate and calcium ions to trigger hydroxyapatite formation. The second cement used was RelyX (RelyX Luting Plus Automix Resin Modified Glass Ionomer Cement; 3M: St. Paul, MN, USA), a radiopaque, fluoride-releasing RMGI cement frequently used in pediatric dentistry and was therefore of interest to compare to the manufacturer’s recommended product. Given the variation in tooth size and shape, along with a passively fitting crown, tooth preparation and cement volume could not be standardized. In order to remove potential confounders, all efforts were made to remove excess cement on the marginal interface before light curing.

Finger pressure was used to properly seat and stabilize crowns for approximately 20 s. Crowns were first flash cured for 5–10 s with a (800–1200 mW/cm^2^) curing light and then for an additional 10 s on facial, lingual and occlusal surfaces mimicking the clinical situation where interproximal sites are not accessible. Following cementation, the crowns were scanned again using CAD technology and the external crown surface area (mm^2^) including marginal cement filling the space between passively fitted crown and the tooth and the outside volume (mm^3^) were calculated using the Meshmixer© software (SCR_015736, Cura v.4.4, Ultimaker, Netherlands) (Figure 1B) All teeth were stored in moist containers for 24–48 h before retrieval was initiated. For each debonding experiment, the same experimental steps were repeated for each group.

### 2.1. Experiments

In the first experiment, prefabricated zirconia crowns were cemented on primary (G1-BC1, N = 12) and permanent (G2-BC1, N = 12) teeth with BioCem cement (BioCem Universal BioActive Cement, NuSmile, Houston, TX, USA). Laser irradiation in the first experiment was performed using the following settings: 4.5 Watts, 15 Hertz, 20 water and 20 air with the Turbo MX9 handpiece (WaterlaseiPlus, Biolase, CA, USA). Following debonding, the crowns and teeth were cleaned of residual cement by first gross removal with a scaler, followed by air abrasion and water rinse to achieve a clean, cement-free tooth and crown surface suitable for recementation. Between experiments, no alterations of the abutment or the crown intaglio surfaces occurred, so the fit of the crowns remained unchanged throughout the experiments.

Crowns were recemented with BioCem cement on primary (G1-BC2, N = 12) and permanent (G2-BC2, N = 12) teeth for the second experiment. The settings for laser irradiation in the second experiment were slightly higher at: 5 Watts, 15 Hertz, 50 water and 50 air with the Turbo MX9 handpiece.

The third experiment involved only permanent teeth (G2-RX1, N = 12) using RelyX, a commonly used RMGI cement (RelyX, 3M ESPE, St. Paul, MN, USA). Following the second debonding, teeth and crowns were cleaned using the same protocol and crowns were recemented on the 12 permanent teeth. Settings for the laser irradiation in the third experiment were: 5 Watts, 15 Hertz, 50 water and 50 air with the Turbo MX9 handpiece.

### 2.2. Laser Debonding Procedure

The crowns were irradiated on the buccal, lingual and occlusal surfaces for 30 s in continuous motion of the handpiece in a back and forth motion 4–5 mm from the crown surface. The interproximal surfaces were not irradiated directly to mimic adjacent teeth being present in the mouth. To attempt the crown removal, the crown retrieval was first attempted by digital manipulation. If crown could not be removed by digital manipulation, gentle tapping forces using a traditional crown removal instrument were applied to the buccal and lingual margins, simulating clinical access to those surfaces. If the crown could not be successfully removed by digital manipulation or tapping, the tooth was subjected to additional 30 s intervals of laser irradiation followed by additional attempts of crown removal after each of the intervals until the crown could be retrieved.

### 2.3. Pulpal Temperature

A 3–4 mm diameter hole was drilled through the furcation into the pulpal chamber of each tooth (Figure 1C) to enable insertion of a temperature probe (Sper Scientific^®^ 800008, Scottsdale, AZ, USA) for measurements of pulpal temperatures. (Figure 1D) Pulpal temperatures were recorded at the baseline and throughout the entire debonding procedure in 30-s intervals.

### 2.4. Scanning Electron Microscopy Analysis

The surfaces of teeth and crowns of one primary and two permanent teeth following first and second laser irradiation experiments were examined under scanning electron microscope (SEM) (JEOL 6610LV, JEOL, Tokyo, Japan) to examine structural integrity and possible surface damage to the crown and tooth due to laser irradiation. Following successful debonding, the underlying intaglio surface and the cameo surface of the crown and the cameo surface of the tooth were inspected and analyzed.

### 2.5. Statistical Methods

Differences in average debond time and average increase in temperature between primary and permanent teeth were assessed using Wilcoxon rank sum tests. Kruskal–Wallis test with Dwass, Steel, Critchlow–Fligner Method post hoc pairwise comparisons was used to determine differences in the debond time (dependent variable) across the primary and permanent teeth groups debonding attempts (independent variables). Pearson’s correlations were used to determine the association between the crown metrics and the time necessary to debond the crown. Multiple linear regression was used to determine the relationship between time to debond and the cement volume and the ratio of the outer to inner surface area. Assumptions of the linear regression model were verified.

## 3. Results

A total of 12 permanent and 12 primary teeth were utilized in this study. Permanent teeth were debonded a total of three times and primary twice. Summary of the volume and surface area metrics for the crowns are given in Table 1.

### 3.1. Debonding Time

The median time for primary molar crown removal using the Er,Cr:YSGG laser was 2 min (IQR: 1:30–2:45). For permanent crowns, the median time for removal was 3 min (IQR: 2:30–6:40). This difference was statistically significant (*p*-value < 0.0001).

Due to the high correlations among the crown metrics, they could not all be considered for the overall models for time to debond. The cement volume and the ratio of inner to outer surface area were selected as the most informative and utilized for the analysis. The pairwise correlation for these two variables was low (r = 0.04, *p*-value = 0.751).

### 3.2. Primary Teeth

The difference in time for debond attempts with primary teeth (G1-BC1, G1-BC2), was not statistically significant (*p*-value = 1.00). The median debond times were 1:45 (IQR: 1:30–2:45) and 2:00 (IQR: 1:15–2:45), respectively. Time to debond crowns from the primary teeth was only significantly correlated with the cement volume (r = 0.62, 0.63 for the two debond attempts respectively). Correlations are given in Table 2. The ratio of the outer to inner surface area was not significantly associated with the debond time for the primary teeth for either the first (*p*-value = 0.635) or the second (*p*-value = 0.606) debond attempts. (Figure 2A) The cement volume was marginally significantly associated with the debond time for primary teeth at both debond attempts. (Figure 2B) For the first attempt, a 1-mm^3^ increase in cement volume had an estimated 0.85 (95% CI: −0.03–1.73) second increase in the time to debond (*p*-value = 0.057). For the second attempt, a 1 mm^3^ increase in cement volume was associated with a 0.72 (95% CI: −0.01–1.45) second increase in the time to debond (*p*-value = 0.052). Complete results are given in Table 3.

### 3.3. Permanent Teeth

For permanent molars, there were significant differences in the debonding times in the three experiments (*p*-value = 0.0010). The first experiment (G2-BC1/Debond 1) was the fastest, with a median of 2:30 (IQR: 1:31–3:00), and was significantly faster than the second experiment (G2-BC2/debond 2) (Median: 3:30, IQR: 3:00–4:30 *p*-value = 0.0140). The third experiment (G2-RX/debond 3) required the longest time (median = 5:00, IQR: 4:00–6:30) and took significantly more time than debond 1 (*p*-value = 0.0033). Debond 3 was also longer than debond 2, but this difference was not statistically significant (*p*-value = 0.2689). All further analyses were performed by debond attempt due to the varying conditions of the attempt.

The time to debond crowns from permanent molars was significantly positively correlated with the inner surface area (r = 0.63), inner volume (r = 0.70), outer volume (r = 0.73), and the cement volume (r = 0.56). Although they were not statistically significant, it was also negatively correlated with the ratio of the outer to the inner surface area (r = −0.44), and the ratio of the volume (r = −0.30). (Figure 2C) This indicates that crowns that are larger relative to the abutment were faster to debond. A similar pattern was seen for the second experiment (debond 2), with significant positive correlations between the time to debond and: total crown volume (r = 0.61), inner surface area (r = 0.77), outer surface area (r = 0.76), inner (r = 0.76) and outer (r = 0.73) volume. Again, the debond time was negatively correlated with the ratio of the inner to outer surface area (r = −0.41) and volume (r = −0.37) but these were not statistically significant. The time for the third experiment, debond 3, which utilized a different cement, was not significantly correlated with any of the measures, but the strongest correlation was with the cement volume (r = 0.38). Correlations are given in Table 2.

Cement volume was associated with a significant increase in the time to debond permanent crowns for the first debond experiment settings (G2-BC1). (Figure 2D) For a 1 mm^3^ increase in the volume of the cement, the debond time required an additional 1.2 s (*p*-value = 0.0369; 95% CI: 0.09–2.28). There was marginal evidence of a significant decrease in first permanent molar debond time by an estimated 16.7 s for a 1-unit increase in the ratio (*p*-value = 0.0853; 95% CI: −2.9–36.3). Cement volume was not significantly associated with permanent molar debond time for the second (G2-BC2) (*p*-value = 0.415) or third (G2-RX) (*p*-value = 0.2485) debond attempts. There was marginal evidence of a decrease in second debond time (Debond 2) based on the ratio, with an estimated 13.23 s decrease for a 1-unit increase in the ratio (*p*-value = 0.195; 95% CI: −8.1–35.6). Complete results are given in Table 3.

### 3.4. Temperature

The mean temperature changes were 2.48 °C (SD = 1.43) for permanent teeth and 3.14 °C (SD = 1.88) for primary teeth. Although the primary teeth had greater temperature change, the difference was not statistically significant (*p* = 0.122). Data on temperature changes are given in Table 4.

### 3.5. Clinical and Scanning Electron Microscopy (SEM) Examination

After debonding, each crown and tooth were examined to analyze the adherence of cement to the dentin or crown. Visual examination of the crown and abutment tooth showed minor cement ablation and no visual cracks or damage to the material or tooth. The SEM analysis was made for one sample from each group after each treatment. The cement appeared to either stay attached to the intaglio surface of the crown (Figure 3A) or tooth surface. (Figure 3B) Neither teeth nor crowns exhibited structural changes or damage suggestive of photoablation or thermal ablation. Some remaining cement particles were noted; however, no notice of carbonization, surface damage, micro cracks, or any other change was visible on the surface of teeth or crowns. Slight partial ablation of the cement caused by Er,Cr:YSGG laser irradiation was occasionally observed on both tooth surface (Figure 3C) and intaglio surface of the crown (Figure 3D) with no visible cracks or fractures on the SEM analysis.

## 4. Discussion

Lasers complement traditional dental practice and can help reduce procedure time, improve treatment outcomes and increase patient acceptance. As a less invasive and a more efficient treatment option, laser-assisted crown removal presents a good alternative to rotary instrumentation making the procedure easier on the patient and less damaging to the tooth structure and restoration [9]. The laser energy is partially absorbed by the remnant water in the resin monomer and then by both the water and inorganic matter present in the dentin. During the thermo-mechanical reaction between laser light and a chromophore in the luting cement, water micro explosions occur and result in reduction of the adhesive resin strength between the tooth and the crown [23,24].

No studies have yet been published which examine the application of lasers to assist crown removal for prefabricated zirconia crowns on primary teeth. Prefabricated zirconia crowns offer improved esthetic appearance in the pediatric population and have been used to restore primary and permanent teeth in pediatric and adolescent population. Occasionally, indications for crown retrieval present themselves including endodontic therapy, recurrent caries and fabrication of a new restoration. Crowns may occasionally also not be optimally seated during the cementation process and this method provides reversible technique to retrieve the crown and improve the crown seat and cementation. Crown removal can be achieved atraumatically using erbium laser irradiation. In this study, greater cement volume was associated with a longer debonding time reinforcing the importance of proper crown size and fit as a better fitting crown requires less cement to fill the space between the abutment and crown. Longer irradiation and more laser energy may possibly be required to ablate the water and monomer components in the larger cement volume which somewhat coincides with the poorer passive “crown-fit”. Given that the crown and cement volumes were not standardized, more research is needed on this topic to further explore the differences.

The use of a tapping force versus just digital manipulation for crown removal was not standardized in this study, but can also affect the crown retrieval time. It is difficult to predict at what point during the laser irradiation, the bond between the crown and tooth is loose enough to enable the crown removal. Thus, following incremental irradiation, attempts to remove the crown range from tapping on the crown margin to just being able to remove the crown digitally from the tooth. The crown may therefore be retrieved following irradiation by tapping or digital manipulation depending on the retentive features of the tooth and the extent and strength of the remaining bond. When stronger tapping forces are applied, the crown can be removed following shorter irradiation as mechanical force is used to break residual bond rendering time shorter as evident from the results in the first set of experiments. Tapping forces on an extracted tooth can often exceed the forces that can be tolerated in-vivo, especially on a pediatric patient. For patients unable to tolerate tapping forces, or when tapping may cause iatrogenic damage to the tooth or restoration, the irradiation time can be extended by 1–2 min enabling the retrieval of the crown by using only digital manipulation.

Laser irradiation can result in increased temperature in the pulpal chamber and surrounding tissues. An increase in pulpal temperature over 5.5 °C could cause irreversible damage to the pulp tissue [25]; while temperature below 30 °C is considered safe for the vitality of the dental pulp [25,26,27,28]. For all experimental groups in this study, the temperature in the pulpal chamber on average increased 2.48 °C for permanent teeth and 3.14 °C for primary teeth, remaining well below the 5.5 °C and 30 °C during the laser irradiation. Small differences observed among the permanent and primary teeth groups can be attributed to the proximity of the pulpal chamber to the irradiated surfaces and the setting of the laser. Primary teeth have larger pulpal chambers and a thinner layer of dentin and thus the heat generated during irradiation may have stronger thermal effects on the pulp. Similar trends were observed in the smaller sized permanent molars. Irradiation at higher laser settings resulted in higher temperature increases but did not improve the debonding efficiency. Therefore, it may be safer to use lower laser settings on teeth with larger pulps and thinner dentinal barrier between the cemented crowns and the pulpal chamber. Longer irradiation time did not lead to higher temperature increases and therefore additional irradiation time may be preferable to aggressive tapping forces or higher power laser settings. Future research may be necessary to investigate the effect on pulpal temperature in anterior teeth with smaller crowns.

Irradiation with Er,Cr:YSGG laser does not seem to damage the retrieved prosthesis or the tooth structure rendering the retrieved restoration reusable when indicated. Following Er,Cr:YSGG irradiation, no structural surface damage to the tooth or crown was observed on SEM analysis. These observations are consistent with previous findings for Er;YAG and Er,Cr:YSGG laser at similar settings used to retrieve zirconia crowns from zirconia implant abutments [21] and teeth [22].

This study is not without limitations. The same teeth were reused for second and third experiments making it difficult to determine the influence of repeated bonding and irradiating. This study lacks a control group, which could consist of traditional rotary crown removal method. Limited evidence exist comparing the rotary to laser assisted crown removal rendering this an interesting topic for future research. Pulpal temperature was affected by the room and water temperature, which had slight daily variations. This study was performed on extracted teeth and therefore does not take into the account the challenges encountered in clinical implementation of this technique. The sample size was small and appropriate pilot data were not available to perform a thorough sample size calculation. Power analysis determined that a sample size of 12 per group would be able to detect an effect size of 0.3 with 80% power. This effect size was deemed sufficient because it would allow for detecting small but not trivial differences, as defined by Cohen [29]. The primary aim of the study was to assess the feasibility of using an Er,Cr:YSGG laser for retrieval of prefabricated zirconia crowns cemented with resin-modified glass ionomer (RMGI) cements from primary and permanent molars, not to demonstrate statistical significance for a particular research question. This study may not have been adequately powered to detect differences in the experiments and results of inferential statistics should be interpreted with caution. However, the study and the sample size were adequate for demonstrating the usefulness and feasibility of an Er,Cr:YSGG laser for retrieval of prefabricated zirconia crowns cemented with resin-modified glass ionomer (RMGI) cements from primary and permanent molars. The presence of all potential confounders could not be controlled due to various size and shape of teeth, individual characteristics of tooth preparation, various fit of prefabricated crowns resulting in a non-standardized volume of cement. Tapping forces also varied based on shape of tooth, accessibility of the crown margin, position of the tapping instrument, retentive features of abutment, fit of the crown and operator strength and dexterity.

This study demonstrated that irradiation with Er,Cr:YSGG laser at the presented settings appears to have little or no effect on prefabricated zirconia crowns and that the crowns can be reused without compromised structural properties following laser assisted retrieval. The laser settings used in the present study appear to be an effective, safe and non-invasive method to remove prefabricated zirconia crowns cemented with RMGI cements from permanent and primary teeth. Crown removal using an Er,Cr:YSGG laser could be applied in clinical practice to remove prefabricated zirconia crowns in pediatric and adolescent patients.

## Figures and Tables

**Figure 1 materials-13-05569-f001:**
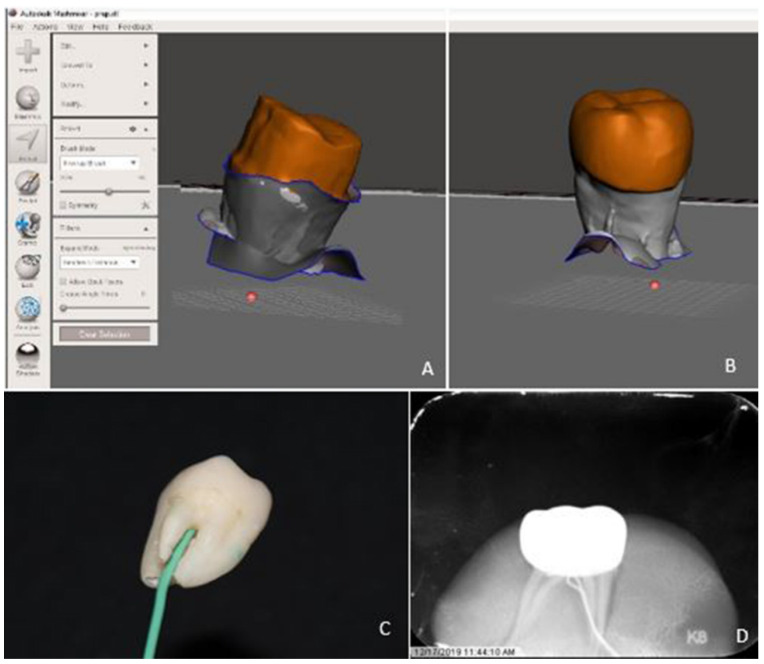
Prepared surfaces of teeth (**A**) and cameo surfaces of the crowns (**B**) were scanned to calculate tooth surface area and the tooth volume. To measure pulpal temperature changes, a 3 mm diameter hole was drilled through the furcation (**C**) to enable insertion of a temperature probe into the pulpal chamber (**D**).

**Figure 2 materials-13-05569-f002:**
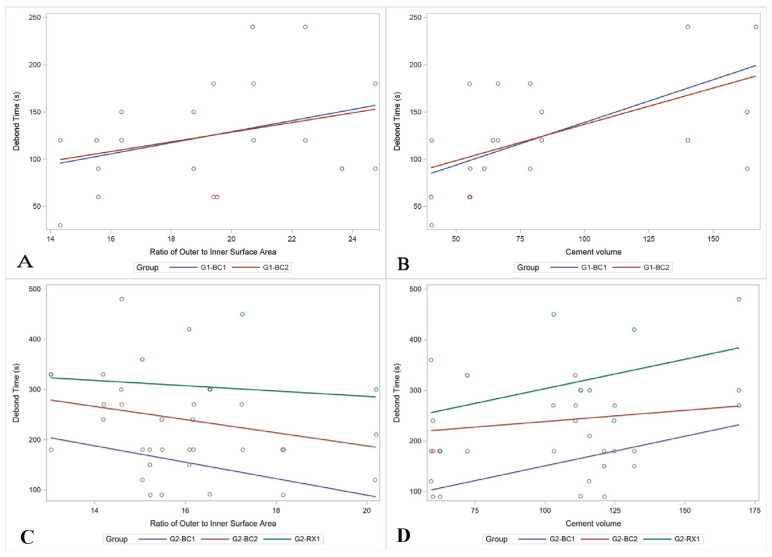
Correlations of debond time with ratio of outer to inner surface area (**A**,**C**) and cement volume (**B**,**D**) for primary (**A**,**B**) and permanent (**C**,**D**) teeth.

**Figure 3 materials-13-05569-f003:**
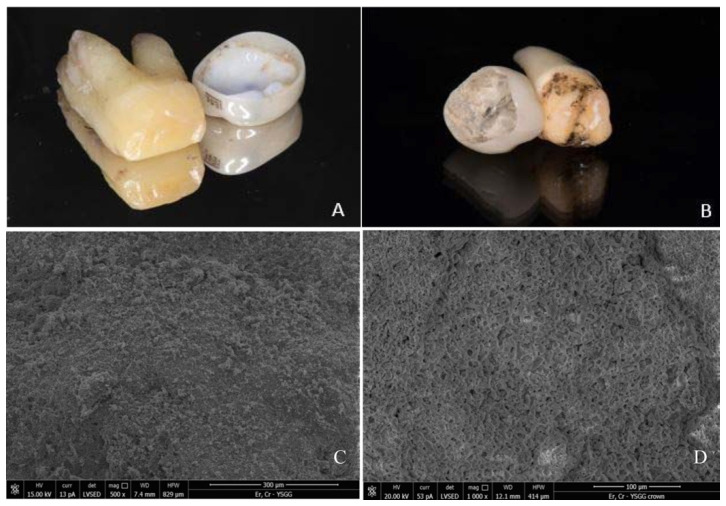
Clinical view of the tooth and crown following laser-debonding from primary (**A**) and permanent (**B**) tooth. Tooth surface (**C**) and intaglio surface of the crown (**D**) appeared to be undamaged and covered by the residual cement on the SEM analysis.

**Table 1 materials-13-05569-t001:** Summary Statistics (mean, SD) for Permanent and Primary Crowns.

	Mean, SD	
	Permanent Teeth (n = 12)	Primary Teeth (n = 12)
Crown Volume	201.76, 32.69	93.92, 43.83
Inner surface area	171.69, 28.16	88.77, 26.85
Outer surface area	270.77, 25.94	166.31, 43.43
Ratio surface area (outer:inner)	160.02, 19.03	193.12, 33.63
Inner volume	233.65, 67.20	76.45, 40.10
Outer volume	537.47, 108.75	254.16, 102.52
Ratio volume (outer:inner)	236.8, 36.21	379.89, 166.57
Cement volume	103.69, 34.01	84.64, 45.67

Note: Volume in mm^3^, Surface Area in mm^2^, Ratios were scaled by 100.

**Table 2 materials-13-05569-t002:** Pairwise correlations between crown metrics and time to debond.

	Permanent Teeth			Primary Teeth	
	Debond 1	Debond 2	Debond 3	Debond 1	Debond 2
Crown Volume	0.433	**0.613 ***	0.070	0.328	0.201
Inner surface area	**0.626 ***	**0.766 ***	0.059	0.225	0.181
Outer surface area	0.506	**0.760 ***	0.008	0.573	0.408
Ratio surface area (outer:inner)	−0.439	−0.406	−0.096	0.298	0.310
Inner volume	**0.697 ***	**0.764 ***	0.219	−0.085	0.188
Outer volume	**0.731 ***	**0.725 ***	0.251	0.381	0.441
Ratio volume (outer:inner)	−0.303	−0.374	−0.091	0.505	0.164
Cement volume	0.561	0.244	0.377	**0.623 ***	**0.632 ***
Temperature Change	0.046	−0.195	−0.125	0.583	0.181

* values in bold are significant *p*-value < 0.05.

**Table 3 materials-13-05569-t003:** Linear regression results for relationship between cement volume, ratio of outer to inner surface area and time to debond.

	Cement Volume		Ratio of Outer:Inner Surface Area	
**Permanent**	**B (SE)**	***p*-Value**	**B (SE)**	***p*-Value**
**Debond 1:** 4.5w, BioCem Cement	1.19 (0.48)	0.0369	−16.71(8.65)	0.0853
**Debond 2:** 5.0w, BioCem Cement	0. 45 (0.53)	0.4150	−13.23 (9.44)	0.1948
**Debond 3:** 5.0w, RelyX Cement, no minimal tapping	1.17 (0.95)	0.2485	−5.63 (16.90)	0.7468
**Primary**	**B (SE)**	***p*-Value**	**B (SE)**	***p*-Value**
**Debond 1:** 4.5w, BioCem Cement	0.85 (0.39)	0.0572	2.60 (5.29)	0.6345
**Debond 2:** 5.0w, BioCem Cement	0.72 (0.32)	0.0524	2.34 (4.38)	0.6055

**Table 4 materials-13-05569-t004:** Summary of crown temperature during debonding.

	Maximum Temperature	SD	Range	Average Delta	SD	Range
Permanent						
Debond 1	22.9	0.93	21.9–25.6	2.4	0.98	0.1–4.5
Debond 2	26.8	1.03	24.5–28.6	2.1	1.19	0.4–4.2
Debond 3	26.6	1.16	25.5–29.9	3.0	1.98	1.6–8.9
Primary						
Debond 1	26.1	2.82	22.1–30.7	4.0	2.17	1.0–8.1
Debond 2	26.7	1.49	24.4–29.9	2.4	1.17	1.1–4.9

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
