# Peer review of "Retrieval of Prefabricated Zirconia Crowns with Er,Cr:YSGG Laser from Primary and Permanent Molars"

_materials, 2020, doi:10.3390/ma13235569_

Round 1
Reviewer 1 Report
The manuscript aimed to assess the feasibility of use of Er,Cr:YSGG laser for removing prefabricated zirconia crowns. I can imagine that the authors did lots of work on this in-vitro study. However, I have some comments on this manuscript:
INTRODUCTION
- Is well written and provide sufficient background. Nevertheless, it would be interesting to add a paragraph about cements, since to types of them were used in the present study.
- Line 50: Add square brackets before and after the reference number.
MATERIALS & METHOD
- In order to avoid potential confounders. Has any effort been made to standardize sample preparation? For example, was a time limit set for preparing the samples? How was passive fit ensured? Was the same amount of cement applied to all samples? etc. Please, take into account that in vitro studies must control all potential confounders to avoid biased results.
- If human samples were used, why was the research exempt from ethical approval?
- In G2-BC1 and G1-BC2 there were 13 teeth, but in G2-RX1 there were only 12. Please, justify.
- Why did the authors use two different types of cement? Is it really relevant? It should be noted that this fact introduces a new covariate.
- Were the samples treated randomly? If not, could this fact have conditioned the results?
- Why did the authors decide not to include a positive control? (i.e. high-speed rotary instruments?
- How did the authors ensure that the fit/performance of the samples was not affected between attempts?
- Please, rephrase lines 148 and 149. Are difficult to understand.
- Due to the limited sample size (<20 samples), the use of parametric tests seems quite suprising. Did the authors verify the assumptions of the models? More specifically, are the authors sure that the assumptions of normality and homocedasticity were met?
- Which covariates were included in the multiple regression model? Was collineality checked?
RESULTS
- General considerations
- Line 160: "A total of 12 permanent and 12 primary teeth were utilized in this study". Weren't they 13 and 12?
- For every mean, please add standard deviation and range.
- For every mean difference test, please add 95% confidence interval.
- For all p-values, reduce the number of decimals from 4 to 3.
- Change mm3 to mm3.
- Primary teeth
- Was the average debond time exactly the same for the two groups? This seems quite surprising.
- Line 182: add p-value.
- Line 183: remove one of the periods.
- Permanent teeth
- Lines 187, 188 and 190: It is easier to understand the results if times are presented as XX minutes and YY seconds. Please, amend.
- Lines 211 and 212: Add =.
- Temperature
- Only general results are presented. Were there differences between the different protocols used?
DISCUSSION
- Line 283: Add references supporting this statement.
- Discuss about the potential confounders (tooth preparation, amount of cement, type of cement, force applied, number of attempts, etc.). Could the authors have made any effort to control them?
- Discuss how the limitations may have affected the results. As any source of bias, they over- or infraestimate the effect of the intervention?
- The main aim of the study was to "analyze the feasibility of use of Er,Cr:YSGG laser for retrieval of prefabricated zirconia crowns cemented with resin-modified glass ionomer (RMGI) cements from primary and permanent molars while also establishing most effective and least aggressive laser settings". According to the obtained results, could the authors establish which is the most effective and safest protocol?
AUTHOR CONTRIBUTIONS
- Please, add.
TABLE 1
- Are the data presented as: mean, standard deviation? Please clarify. This seems confusing.
FIGURE 2
- Looking at the results it seems difficult to assume normality and homocedasticity assumptions. Please, check.
Author Response
Reviewer 1:
Comments and Suggestions for Authors
The manuscript aimed to assess the feasibility of use of Er,Cr:YSGG laser for removing prefabricated zirconia crowns. I can imagine that the authors did lots of work on this in-vitro study. However I have some comments on this manuscript:
INTRODUCTION
- Is well written and provide sufficient background. Nevertheless, it would be interesting to add a paragraph about cements, since to types of them were used in the present study.
A paragraph about cements has been added to Materials and Methods.
Glass ionomer cements have long been the standard for stainless steel crown cementation. Along with fluoride release, glass ionomer cements also bond to dentin. Unlike the “snap fit” of a stainless steel crown, zirconia crowns require a passive fit, possibly leading to a more “open” and “compromised” marginal interface, placing great emphasis on cement composition. Resin-modified glass ionomers (RMGI) combine the properties of traditional glass ionomers with the handling and light-curing abilities of resin, culminating in increased retention and longevity. Two cements were used in this study. One was the manufacturer’s proprietary cement, NuSmile BioCem Universal Bioactive Cement. A RMGI with additional bioactive properties that claim to release fluoride, phosphate and calcium that trigger hydroxyapatite formation (RelyX Luting Plus Automix Resin Modified Glass Ionomer Cement; 3M: St. Paul, MN, 2011). The second RMGI used was RelyX Luting Plus Automix. A radiopaque, fluoride-releasing, resin-modified glass ionomer commonly used in pediatric dentistry. (BioCem Universal BioActive Cement; NuSmile: Houston, TX).
- Line 50: Add square brackets before and after the reference number.
Square brackets have been added.
MATERIALS & METHOD
- In order to avoid potential confounders. Has any effort been made to standardize sample preparation? For example, was a time limit set for preparing the samples? How was passive fit ensured? Was the same amount of cement applied to all samples? etc. Please, take into account that in vitro studies must control all potential confounders to avoid biased results.
Text was added to Materials and Methods to address this comment.
Preparing the samples largely depends on the size and shape of the tooth, so standardization was not possible as teeth varied in shape and size. Passive fit was determined digitally after crown preparation and before cementation. Given the variation in tooth size and shape, along with a passively fitting crown, preparation of samples and cement volume could not be standardized. In order to remove potential confounders, all efforts were made to remove excess cement before light curing.
Additionally, these potential confounders were discussed in the results and conclusions section when crown and cement volumes were compared between teeth.
- If human samples were used, why was the research exempt from ethical approval?
The project was reviewed by the Institutional Review Board and deemed to be Not Human Subjects Research because the human teeth were stripped of all identifiers and could not be tied to any identifiable data.
- In G2-BC1 and G1-BC2 there were 13 teeth, but in G2-RX1 there were only 12. Please, justify.
12 teeth were used in all groups and the text has been corrected.
- Why did the authors use two different types of cement? Is it really relevant? It should be noted that this fact introduces a new covariate.
One type of cement was used as manufacturer recommends it to be used with their prefabricated zirconia crowns. The second cement was added to represent the most commonly used cement in pediatric practice to cement the prefabricated zirconia crowns.
This explanation was added to Materials and Methods:
“Two cements were used in this study. One was the manufacturer’s proprietary cement, NuSmile BioCem Universal Bioactive Cement. A RMGI with additional bioactive properties that claim to release fluoride, phosphate and calcium that trigger hydroxyapatite formation (RelyX Luting Plus Automix Resin Modified Glass Ionomer Cement; 3M: St. Paul, MN, 2011). The second RMGI used was RelyX Luting Plus Automix. A radiopaque, fluoride-releasing, resin-modified glass ionomer commonly used in pediatric dentistry. (BioCem Universal BioActive Cement; NuSmile: Houston, TX).”
- Were the samples treated randomly? If not, could this fact have conditioned the results?
All samples were treated in the same order: BC1, BC2, RX1 (for permanent teeth only). Therefore, it’s possible that there was carryover effect from each bonding/debonding.
- Why did the authors decide not to include a positive control? (i.e. high-speed rotary instruments?
The study did not include a high speed crown sectioning as crowns and most teeth would be destroyed after applying this retrieval method. Aim of this study was to explore laser assisted retrieval under various settings and different cements.
Data exists on high-speed rotary instruments facilitated crown removals.
- How did the authors ensure that the fit/performance of the samples was not affected between attempts?
There were no alterations on the abutment surface or crown intaglio surface or shape, so fit remained unchanged throughout the experiments. The text was added to the Materials and Methods:
“Between experiments, no alterations on the abutment or the crown intaglio surfaces occurred, so the fit of the crowns remained unchanged throughout the experiments.”
- Please, rephrase lines 148 and 149. Are difficult to understand.
Text has been rephrased.
- Due to the limited sample size (<20 samples), the use of parametric tests seems quite suprising. Did the authors verify the assumptions of the models? More specifically, are the authors sure that the assumptions of normality and homocedasticity were met?
Some of the variables did violate the assumptions of normality however the analyses were re-run with nonparametric alternatives (analysis on the ranks) and the results yielded the same results in terms of statistical significance. We have reported on mean differences for the sake of interpretability by the audience. Unequal variance methods were utilized where appropriate. We have addressed this in the discussion section regarding the study limitations.
- Which covariates were included in the multiple regression model? Was collineality checked?
As stated in 3.1: Due to the high correlations among the crown metrics, they could not all be considered for the overall models for time to debond. The cement volume and the ratio of inner to outer surface area were selected as the most informative and utilized for the analysis. The pairwise correlation for these two variables was low (r=0.04, p-value=0.7513).
RESULTS
- General considerations
- Line 160: "A total of 12 permanent and 12 primary teeth were utilized in this study". Weren't they 13 and 12?
12 teeth were used in all groups and the text has been corrected.
- For every mean, please add standard deviation and range.
These have been added to the text
- For every mean difference test, please add 95% confidence interval.
These have been added to the text
- For all p-values, reduce the number of decimals from 4 to 3.
All p-values have been shortened to 3 decimal places.
- Change mm3 to mm3.
It has been corrected.
Primary teeth
- Was the average debond time exactly the same for the two groups? This seems quite surprising.
Yes, the debond was the same for the two primary teeth groups. The variance was not the same but the average was. Mean, SD for each group of primary teeth: G1-BC1: 125, 66.3 and G1-BC2: 125, 55.5
- Line 182: add p-value.
p-value has been added.
- Line 183: remove one of the periods.
One period has been removed.
Permanent teeth
- Lines 187, 188 and 190: It is easier to understand the results if times are presented as XX minutes and YY seconds. Please, amend.
We have reformatted into minutes and seconds.
- Lines 211 and 212: Add =.
= has been added
Temperature
- Only general results are presented. Were there differences between the different protocols used?
Table 4 contains the temperature analysis for all experiments. The observations regarding temperature changes are referred to in the results section and discussed in the discussion:
RESULTS:
“The mean temperature changes were 2.48℃ (SD=1.43) for permanent teeth and 3.14℃ (SD=1.88) for primary teeth. Although the primary teeth had greater temperature change, the difference was not statistically significant (p=0.122). Data on temperature changes are given in Table 4.
DISCUSSION:
“For all experimental groups in this study, the temperature in the pulpal chamber remained within tolerable range during the laser irradiation. Small differences observed among the groups can be attributed to the proximity of the pulpal chamber to the irradiated surfaces and the setting of the laser. Primary teeth have larger pulpal chambers and thinner layer of dentin and thus the heat generated during irradiation may have stronger thermal effects on the pulp. Similar trends were observed in the smaller sized permanent molars. Irradiation at higher laser settings did not improve the debonding efficiency, but did result in higher temperature increases. Therefore, it may be safer to use lower laser settings on teeth with larger pulps and thinner dentinal barrier between the cemented crowns and the pulpal chamber. Longer irradiation time also did not lead to higher temperature increases and therefore additional irradiation time may be preferable to aggressive tapping forces or higher power laser settings.”
DISCUSSION
- Line 283: Add references supporting this statement.
Several references have been added along with additional text.
- Discuss about the potential confounders (tooth preparation, amount of cement, type of cement, force applied, number of attempts, etc.). Could the authors have made any effort to control them?
The authors have expanded on this in Materials and methods, results and discussion.
- Discuss how the limitations may have affected the results. As any source of bias, they over- or infraestimate the effect of the intervention?
Not sure I understand this point. Limitations have been considered in the discussion.
- The main aim of the study was to "analyze the feasibility of use of Er,Cr:YSGG laser for retrieval of prefabricated zirconia crowns cemented with resin-modified glass ionomer (RMGI) cements from primary and permanent molars while also establishing most effective and least aggressive laser settings". According to the obtained results, could the authors establish which is the most effective and safest protocol?
Authors considered time required for removal as a measure of effectiveness and lower laser settings as safer than higher settings. Based on these two outcomes, we have speculated that lower settings present a safer option since higher setting added very little effect to the efficiency.
AUTHOR CONTRIBUTIONS
- Please, add.
Author contributions have been added.
TABLE 1
- Are the data presented as: mean, standard deviation? Please clarify. This seems confusing.
As indicated in the top of Table 1, data presented is Mean, SD. This has also been added to the table caption
FIGURE 2
- Looking at the results it seems difficult to assume normality and homocedasticity assumptions. Please, check.
See comments above regarding departure from normality.

Reviewer 2 Report
The current manuscript investigated the retrieval of prefabricated zirconia crowns with Er,Cr:YSGG laser from primary and permanent molars. The subject might be of interest to the readers, but before considering it for publication it has to be corrected extensively. Followed are my recommendations:
Introduction
As it stands the authors claims as if Er,Cr:YSGG laser was not previously investigated to retrieve zirconia crowns at all, but only lithium disilicate crowns. This is not true. Please add the following references which described it: Exploring the use of pulsed erbium lasers to retrieve a zirconia crown from a zirconia implant abutment by A. Elkharashi et al, Plos One, 2020 and Rechmann P et al. Laser all-ceramic crown removal-a laboratory proof-of-principle study-phase 2 crown debonding time. Lasers Surg Med. 2014. Subsequently focus the introduction on the retrieval of prefabricated zirconia crowns in pedodontic patients.
Materials and methods
- Ethics approval by the university committee is lacking, please add, including reference number.
- Describe the origin of the teeth recruited to the study.
- Describe the burs utilized for the preparation. If the 20-30% of reduction was not measured, omit the sentence.
- The type of cement should be mentioned in the second paragraph, following the sentence: 'dried and cemented on teeth…..'
- Explain in detail how the remnants of cement were removed after retrieval. To my knowledge it can be done only by sandblasting?!
- A severe limitation to the study is that teeth were not placed in typodont model with adjacent mounted natural teeth. This is the only way that one can guarantee that the interproximal surfaces were not irradiated directly in order to mimic adjacent teeth being present in the mouth. Please comment on that.
- Describe (including photograph) the tapping instrument.
- Describe in the statistics, what the dependent are and independent variables of the ANOVA.
Results
- Check the ratios numbers in Table 1.
- Omit the words significantly different from 0 at the legend of Table 2. Keep: significant (p<0.05).
- Omit figure 2. It does not contribute further knowledge to Table 2. Accordingly omit the description referring to that figure.
- The third aim of the manuscript was actually not investigated: 'Additionally, it was of interest to examine how laser irradiation affects structure of the crown and tooth surface'. In order to investigate structure several spectroscopic techniques should be applied (for example: SEM/EDS). This was not done, so I suggest rephrasing, according to what really was done.
Discussion
- It is claimed that the first two experiments on permanent teeth suggested that larger crowns relative to the size of the abutment tooth are faster to debond, but as the correlation between crown volume and time to debond (Table 2) is 0.43 and 0.61 for debond 1 and 2 respectively, the interpretation is mistaken- larger crowns yield longer retrieval times. Moreover, larger crowns should have greater cement amount, and greater cement amount yield longer debonding times, as stated. Please clarify the apparent contradiction.
- The temperature changed in the pulp chamber should be compared to similar studies. For example: Deeb JG et al. Using Er:YAG laser to remove lithium disilicate crowns from zirconia implant abutments: An in vitro study. PLoS One. 2019.
- Enlarge on the limitations of the study and the difficulties in extrapolating to clinical settings.
Author Response
Reviewer 2:
Comments and Suggestions for Authors
The current manuscript investigated the retrieval of prefabricated zirconia crowns with Er,Cr:YSGG laser from primary and permanent molars. The subject might be of interest to the readers, but before considering it for publication it has to be corrected extensively. Followed are my recommendations:
Introduction
As it stands the authors claims as if Er,Cr:YSGG laser was not previously investigated to retrieve zirconia crowns at all, but only lithium disilicate crowns. This is not true. Please add the following references which described it: Exploring the use of pulsed erbium lasers to retrieve a zirconia crown from a zirconia implant abutment by A. Elkharashi et al, Plos One, 2020 and Rechmann P et al. Laser all-ceramic crown removal-a laboratory proof-of-principle study-phase 2 crown debonding time. Lasers Surg Med. 2014. Subsequently focus the introduction on the retrieval of prefabricated zirconia crowns in pedodontic patients.
The statement was revised to reflect only primary teeth. Authors claim that Er,Cr:YSGG laser has not been previously investigated to retrieve prefabricated zirconia crowns from primary teeth.
Both referenced studies are listed in references.
Materials and methods
- Ethics approval by the university committee is lacking, please add, including reference number.
This study was deemed “Not Human Subjects Research” by the VCU IRB. Reference number: HM20019893
- Describe the origin of the teeth recruited to the study.
The origin has been added:
Extracted teeth were collected and stored in saline from patients who had prior treatment planned extractions in both VCU Pediatric Dental and Oral Surgery Clinics.
- Describe the burs utilized for the preparation. If the 20-30% of reduction was not measured, omit the sentence.
The bur information was added:
“A 368-023 football-shaped coarse diamond bur was used for occlusal reduction (Henry Schein®), the interproximal sites and entire clinical crowns of the teeth were reduced using a 169L taper fissure plain carbide bur (Henry Schein®), to establish a chamfer/feather-edge margin the 169L and an 850-010 needle-diamond bur (Henry Schein®) were utilized. Lastly, all the line angles of the preparations were rounded with the needle-diamond bur and football-shaped coarse diamond bur to remove any sharp angles and provide for a slightly tapered preparation that would allow for the zirconia crown to fir passively.”
The statement on the amount of reduction occlusal and circumferentially provides a guideline by the crown manufacturer on how much tooth reduction should be achieved to accommodate the prefabricated zirconia crowns. The term “approximately” has been added to the statement to reflect the non-standardized nature of this reduction.
- The type of cement should be mentioned in the second paragraph, following the sentence: 'dried and cemented on teeth…..'
Information regarding cement types was added to Materials and Methods.
- Explain in detail how the remnants of cement were removed after retrieval. To my knowledge, it can be done only by sandblasting?!
The following text has been added to Materials and Methods:
“After retrieval, and before recementation, cement was grossly removed from the crown’s intaglio surface with a scaler and then removed in its entirety by sandblasting. This process provided a clean, cement-free crown surface for recementation.”
- A severe limitation to the study is that teeth were not placed in typodont model with adjacent mounted natural teeth. This is the only way that one can guarantee that the interproximal surfaces were not irradiated directly in order to mimic adjacent teeth being present in the mouth. Please comment on that.
Teeth were mounted into the typodont (picture added) at first, however taking the teeth in and out changed their position slightly and resulted in the movement of the thermometer and the protocol was thus modified.
- Describe (including photograph) the tapping instrument.
The text has been rephrased to: “gentle tapping forces using a traditional crown removal instrument were applied to the buccal and lingual margins, simulating clinical access to those surfaces.”
- Describe in the statistics, what the dependent are and independent variables of the ANOVA.
This has been added to the methods.
Results
- Check the ratios numbers in Table 1.
The ratios in the table are correct but we have added a footnote to indicate that we multiplied the ratios by 100 to increase interpretability and to keep them on a similar scale as the other variables
- Omit the words significantly different from 0 at the legend of Table 2. Keep: significant (p<0.05).
This has been updated
- Omit figure 2. It does not contribute further knowledge to Table 2. Accordingly omit the description referring to that figure.
We would like to keep Figure 2 since it adds a visual display of the results.
- The third aim of the manuscript was actually not investigated: 'Additionally, it was of interest to examine how laser irradiation affects structure of the crown and tooth surface'. In order to investigate structure several spectroscopic techniques should be applied (for example: SEM/EDS). This was not done, so I suggest rephrasing, according to what really was done.
We already have a short paragraph and two images referring to the findings. We have further expanded the text, which is attached below:
3.5. Clinical and Scanning Electron Microscopy (SEM) Examination
After debonding, each crown and tooth were examined to analyze the adherence of cement to the dentin or crown. Visual examination of the crown and abutment tooth showed minor cement ablation and no visual cracks or damage to the material or tooth. The SEM analysis was made for one sample from each group after each treatment. The cement appeared to stay attached either to the intaglio surface of the crown (Figure 3 A) or tooth surface. (Figure 3B) Neither teeth nor crowns exhibited structural changes or damage suggestive of photo ablation or thermal ablation. Some remaining cement particles were noted; however, no notice of carbonization, surface damage, micro cracks, or any other change was visible on the surface of teeth or crowns. Slight partial ablation of the cement caused by Er,Cr:YSGG laser irradiation was occasionally observed on tooth surface (Figure 3C) and intaglio surface of the crown (Figure 3 D) with no visible cracks or fractures on the SEM analysis.
3.5. Clinical and Scanning Electron Microscopy (SEM) Examination
After debonding, each crown and tooth were examined to analyze the adherence of cement to the dentin or crown. The cement appeared to stay attached either to the intaglio surface of the crown (Figure 3 A) or tooth surface. (Figure 3B) Neither teeth nor crowns exhibited structural changes or damage suggestive of photoablation or thermal ablation. No carbonization was observed on the tooth or crown. Slight partial ablation of the cement caused by Er,Cr:YSGG laser irradiation was occasionally observed on tooth surface (Figure 3C) and intaglio surface of the crown (Figure 3 D) with no visible cracks or fractures on the SEM analysis.
Discussion
- It is claimed that the first two experiments on permanent teeth suggested that larger crowns relative to the size of the abutment tooth are faster to debond, but as the correlation between crown volume and time to debond (Table 2) is 0.43 and 0.61 for debond 1 and 2 respectively, the interpretation is mistaken- larger crowns yield longer retrieval times. Moreover, larger crowns should have greater cement amount, and greater cement amount yield longer debonding times, as stated. Please clarify the apparent contradiction.
Due to small correlation and confusion created by this statement, we have omitted this statement and changed the text in the discussion to discuss the cement volume as it related to the better fit of the prefabricated crown.
CLARIFICATION:
The results indicate that while there is a positive correlation between crown size and debond time, the association between the RATIO of the crown size to its abutment is inversely related to the debond time. So as the crown size gets larger relative to the abutment the debond time decreases. As can be seen in Table 2, the correlation between both the ratio of SA and volume are negatively associated with the debond time (-0.439 and -0.303 for debond 1, for example).
- The temperature changed in the pulp chamber should be compared to similar studies. For example: Deeb JG et al. Using Er:YAG laser to remove lithium disilicate crowns from zirconia implant abutments: An in vitro study. PLoS One. 2019.
The study referenced above was done on implants. Temperature changes were measure on the surface of the implant body and abutment which is not the most optimal comparison to the internal temperature inside the pulp chamber.
A new reference of a recently published study with comparable pulpal temperature assessment has been added.
- Enlarge on the limitations of the study and the difficulties in extrapolating to clinical settings.
The statement was revised to reflect only primary teeth. Authors claim that Er,Cr:YSGG laser has not been previously investigated to retrieve zirconia crowns from primary teeth.

Reviewer 3 Report
Dear authors, thank you for sending this work for consideration. I have found some drawbacks that need to be clarified before your paper can be considered for publication.
Introduction:
Page 2 line 50: please check the number "6" which is not between brackets. Is this a citation?
Materials and Methods:
Page 3 line 96. Following manufacturer's instructions? Which manufacturer?
Why did you test 2 cements for decidous teeth but 3 for permanent?
Results:
Line 160: 12 and 12? Please check this, you say 12 and 13 in materials and methods section.
Did you run a sample size analysis?
You need a group where you test, using your same methods, the traditional techniques (high speed), cannot rely on other studies due to different designs.
Is temperature increase correlated with removal time?
As you mentioned, oral conditions are different so removal of crowns can be more difficult and slower. (for the fact of having adjacent teeth and many other conditions)
So, if the removal time is higher in the mouth, would this mean a greater increase of the pulp temperature?
Do you consider this increase "safe"? Please provide reference.
Conclusions:
Please revise conclusions. They should answer all your objectives
Author Response
Reviewer 3:
Comments and Suggestions for Authors
Dear authors, thank you for sending this work for consideration. I have found some drawbacks that need to be clarified before your paper can be considered for publication.
Introduction:
Page 2 line 50: please check the number "6" which is not between brackets. Is this a citation?
The brackets have been added.
Materials and Methods:
Page 3 line 96. Following manufacturer's instructions? Which manufacturer?
The manufacturer has been referenced following that statement.
Why did you test 2 cements for decidous teeth but 3 for permanent?
One type of cement was used as manufacturer recommends it to be used with their prefabricated zirconia crowns. The second cement was added to represent the most commonly used cement in pediatric practice to cement the prefabricated zirconia crowns.
Results:
Line 160: 12 and 12? Please check this, you say 12 and 13 in materials and methods section.
12 teeth were used in all groups and the text has been corrected.
Did you run a sample size analysis?
The sample size was selected based on resources available but verified using a power calculation, which determined that a sample size of 12 per group would be able to detect an effect size of at least 0.3 with 80% power, which is between a small effect (0.2) and a medium effect (0.5) as defined by Cohen.
You need a group where you test, using your same methods, the traditional techniques (high speed), cannot rely on other studies due to different designs.
This is the limitation and should have been added to the study.
Is temperature increase correlated with removal time?
Temperature change was added to Table 2 to demonstrate that the pairwise correlation between debond time and temp change was not significantly correlated with any of the 5 debond groups.
As you mentioned, oral conditions are different so removal of crowns can be more difficult and slower. (for the fact of having adjacent teeth and many other conditions) So, if the removal time is higher in the mouth, would this mean a greater increase of the pulp temperature? Do you consider this increase "safe"? Please provide reference.
The statement is discussion already exists in regards to this application:
“Longer irradiation time also did not lead to higher temperature increases and therefore additional irradiation time may be preferable to aggressive tapping forces or higher power laser settings. Future research may be necessary to investigate the effect on pulpal temperature in anterior teeth with smaller crowns.”
Conclusions:
Please revise conclusions. They should answer all your objectives
Conclusions at the end of the discussion have been revised.

Round 2
Reviewer 1 Report
The authors have made a considerable effort to resolve the weaknesses detected in the first review.A new author has been added. However, I must insist that if the assumptions are not met, the descriptive analyses have to be based on orders -median and interquartile range-. Similarly, inferential analyses have to be performed using non-parametric tests. Therefore, in my opinion, the current analysis can be considered inappropriate.
Why did the authors consider an effect measure of 0.3 clinically relevant?
Author Response
Comments of Academic Edtor
The study is interestingly elaborating on a clinical problem that needs evidence and supportive research.
The authors would like to thank the reviewers for their generous and constructive feedback. We have made significant effort to address their comments and imporve the manuscript by implementing the recommendations.
Additionally, no new authors were added after first revision of the manuscript, only the order of authros was changed based on the credits and contributions which were added and warranted further reflection following Revision 1.
Following comments are here to address:
1. The authors mention in the introduction of primary teeth and pediatric dental applications but this is a problem not only in children but also in adults. This should be revised accordingly.
2. is there any power analysis calculation for n=12?
Response: Yes, the power analysis determined that a sample size of 12 per group would have 80% power to detect an effect size of 0.3, which is a moderate to small effect size as defined by Cohen and tehrefore should be adequate for detecting clinically meaningful differences. This was included in previous versions in section 2.5 statistical methods. We have highlighted the statement for ease of reviewing.
- What was the IRB protocol number?
Response: The IRB protocol number (HM20019893) has been added to the manuscript in Section 2 Materials and Methods.
- The temperature increase and the association with the pulp vitality are fundamental. Further discussion is here required.
The authors expanded the discussion segment where temperature changes and their relevance is discussed. The temperature changes in all our experimental groups remained well bellow ccritical increases associated with pulp morbidity
5. Linguistic changes are also required.
We have reviewed and made edits to increase the readability of the manuscript.
Reviewer 1, REV 2:
The authors have made a considerable effort to resolve the weaknesses detected in the first review. A new author has been added. However, I must insist that if the assumptions are not met, the descriptive analyses have to be based on orders -median and interquartile range-. Similarly, inferential analyses have to be performed using non-parametric tests. Therefore, in my opinion, the current analysis can be considered inappropriate.
We have updated some of the analyses to nonparametric methods. The assumptions for multiple linear regression were assessed, all were met, and therefore those results were not updated. The relationship was linear (Figure 2), multicollinearity was assessed for cement volume and ratio of surface area (r=0.04, p-value=0.751), the residuals were normally distributed (q-q plot), and there were no apparent patterns in the residuals. Despite this, the sample size is small and the methods may still not be appropriate. We have addressed this in the limitations.
Why did the authors consider an effect measure of 0.3 clinically relevant?
Cohen defined effect sizes as follows: trivial: <0.2, small: 0.2-0.5, moderate: 0.5-0.8, and large: 0.8+. We felt that an effect size of 0.3 would be clinically relevant because it would demonstrate a small but not trivial difference. This description has been included and moved to the limitations section of the manuscript. The primary aim of the study was not to find statistically significant differences in the experiments, but rather to demonstrate the feasibility of the laser for debonding therefore the study may not have been adequately powered to detect differences. But the study and the sample size were adequate for demonstrating the usefulness and feasibility of debonding with the laser.
Reference: Cohen J. Statistical power analysis for the behavioral sciences. 2nd ed. New Jersey: Lawrence Eribaum; 1988.
Reviewer 2 Report
the recommendations were satisfactory addressed
Author Response
Reviewer 2, REV 2:
The recommendations were satisfactory addressed
The authors would like to thank reviewer 2 for his feedback and favorable review of the manuscript.
Reviewer 3 Report
Dear authors, thank you for making some of the requested changes.
I still would not recommend your paper for publication at its present form.
- Your study lacks control group which should be added and results reinterpreted
- My comment about the different cements used (2 for decidious and 3 for permanent) was not answered.
- Regarding my comment about the correlation removal time/temperature, this was not answered, I can't see this information in the table you mention
Author Response
Reviewer 3, REV 2:
Dear authors, thank you for making some of the requested changes.
I still would not recommend your paper for publication at its present form.
- Your study lacks control group which should be added and results reinterpreted
The authors aknowledged this as a limitation of the study in the diuscussion.
This comment however offers a great suggestion for a future study to compare laser assisted crown debonding to rotary crown removal. Majority of the laser debonding studies published to date consist of just experimental (laser-assisted debonding) groups and focus on establishing optimal laser parameters, efficiency and tempertaure changes. A study of comparing the two techniques offers a great opportunity for future research.
- My comment about the different cements used (2 for deciduous and 3 for permanent) was not answered.
The authors expanded on the rationalle of using the second cement in Material and Methods (line 116-121). While the first cement is a manufacturer’s recommended cement, in clinical practice the second cement is a more commonly used and more ubiquitpusly available alternative and was thus of interest to explore its behavior for this application.
- Regarding my comment about the correlation removal time/temperature, this was not answered, I can't see this information in the table you mention
We have highlighted this in the updated manuscript. It is provided in Table 2, last row.